# PROMPT-SSLC: A UNIFIED FRAMEWORK FOR DUAL PROMPT-AUGMENTED SEMI-SUPERVISED SEQUENTIAL LEADER CLUSTERING IN ON-THE-FLY CATEGORY DISCOVERY

## ABSTRACT

On-the-fly Category Discovery (OCD) enables intelligent systems to perform real-time predictions while adapting to emerging categories in dynamic environments. We present Prompt-SSLC, a unified framework that integrates three synergistic components to balance stability and adaptability in streaming data scenarios. First, Semi-Supervised Sequential Leader Clustering (SSLC) dynamically updates prototypes to accommodate incoming data streams, ensuring flexibility in clustering. To enhance discriminability and mitigate prototype overlap, SSLC incorporates a Distance-Aware (DA) update mechanism that optimizes prototype distributions, maintaining inter-class separation as new data arrive. Second, dual prompting augments the foundation model: a *Task Prompt* guides category discovery, while an *Instance Prompt* dynamically recalibrates features to prevent drift toward previously learned classes without requiring retraining. Third, an Open-Set-Aware (OSA) classifier employs uncertainty estimation to identify and filter ambiguous samples, ensuring robust prototype updates. This cohesive integration of streaming clustering, feature recalibration, and uncertainty-aware filtering establishes a robust framework for OCD. Extensive experiments on generic and fine-grained benchmarks demonstrate that Prompt-SSLC achieves significant performance improvements, setting a new state-of-the-art for OCD.

## 1 INTRODUCTION

While intelligent systems have made substantial progress in closed-world paradigms – such as image recognition, where testing data align with predefined training categories – open-world learning, which requires models to handle unseen categories or unfamiliar scenarios, has yet to achieve comparable success. Category discovery has emerged as an intriguing open-world learning problem, garnering increasing attention. The problem has initially been studied as Novel Category Discovery(NCD) (Han et al., 2019; 2021; Zhao & Han, 2021; Jia et al., 2021; Fini et al., 2021) to categorize unlabeled data from novel categories by transferring knowledge from labeled data, which has subsequently been extended to Generalized Category Discovery (GCD) (Vaze et al., 2022; Zhang et al., 2023; Wen et al., 2023; Vaze et al., 2023; Hao et al., 2024; Wang et al., 2024c; Lin et al., 2024; Zhao et al., 2023; Wang et al., 2024b; Liu & Han, 2025; Liu et al., 2025; Han et al., 2024) to address scenarios where unlabeled data come from both known and novel classes. Recently, Continual Category Discovery (CCD) (Zhang et al., 2022; Cendra et al., 2024) is introduced to handle unlabeled data in an incremental learning setting.

Despite encouraging progresses, there are three overlooked issues in the previous category discovery studies: **(i)** the model needs to operate on a collection of unlabeled data each time, but in practice, the model may face a single or a varying number of unlabeled instances over time; **(ii)** the model needs to be optimized each time when it encounters unlabeled samples; **(iii)** there is a lack of mechanism in separating the novel classes from the known ones in the unlabeled data. Very recently, On-the-fly Category Discovery (OCD) introduces a paradigm for real-time category discovery on unlabeled streaming data (see fig. 1), addressing critical gaps between static and dynamic environments. The framework eliminates batch data collection requirements, autonomously detects emerging categories during continuous operation, and maintains predictive performance without iterative model retraining – establishing a flexible solution for evolving data landscapes. As an emerging challenge, there is a sparse literature on studying OCD, with only two existing methods trying to address the

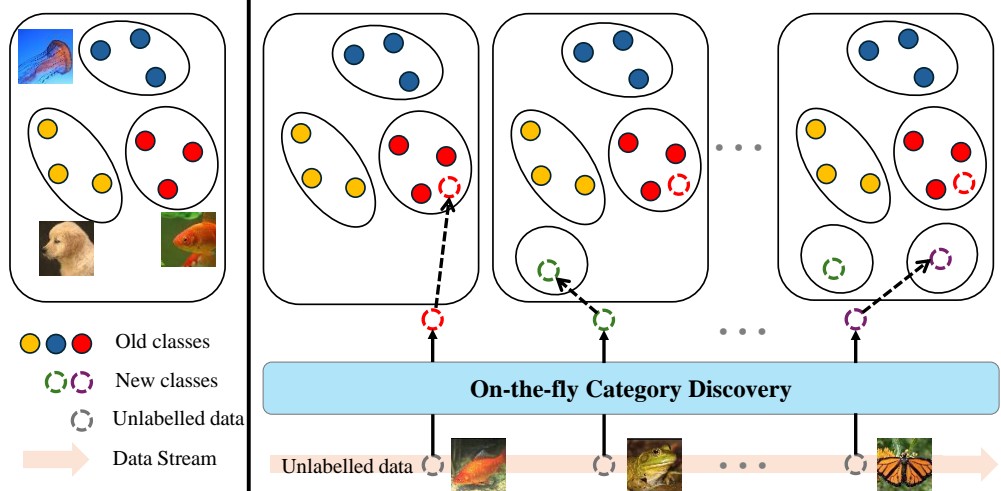

Figure 1: **On-the-fly Category Discovery (OCD).** (Left) During training, the model only has access to labeled data. (Right) During inference, it processes individual instances in real time, dynamically assigning them to existing categories or creating new ones as needed – without requiring access to the full unlabeled dataset.

task, namely SMILE (Du et al., 2023) and PHE (Zheng et al., 2025). SMILE Du et al. (2023) operates by projecting data into a low-dimensional hash space through supervised training on labeled datasets, where matching hash codes define cluster associations. During inference, novel categories emerge as previously unobserved hash code formations in this embedded space. PHE Zheng et al. (2025) advances this paradigm by decoupling performance from dimensionality of hash space while preserving inter-class discrimination. However, there are two major limiations in hash-based approaches: 1) finite capacity constrained by the pre-defined hashing space, limiting scalability when the continuous data stream keeps growing, and 2) critical dependence on meticulously optimized dimensionality parameters. These methods further exhibit static architectural limitations – their hash spaces derive exclusively from initial labeled data without mechanisms for incremental adaptation to streaming inputs.

We propose a novel OCD framework, named Prompt-SSLC, that addresses the aforementioned critical challenges. First, inspired by Sequential Leader Clustering (SLC) (Hartigan, 1973) – a method enabling incremental clustering with dynamically adjustable prototypes – we propose a semi-supervised counterpart tailored for open-category detection (OCD), called Semi-Supervised Sequential Leader Clustering (SSLC). SSLC is a prototype-driven approach that dynamically identifies novel categories in streaming data by comparing incoming patterns to learned class prototypes. This allows continuous adaptation without retraining while preserving constraints from labeled data. In the SSLC module of Prompt-SSLC, we further employs a Distance-Aware (DA) prototype distribution update scheme to enhance robustness by mitigating noise sensitivity and refining prototype separability, ensuring stable performance in evolving data streams. Second, in Prompt-SSLC, to enjoy the benefits of the strong representation by vision foundation models (*i.e.*, DINOv2 (Oquab et al., 2023) in our implementation) for the OCD task, we propose a dual-prompt design, including a *Task Prompt*, which guides category discovery, and an *Instance Prompt*, which dynamically adjusts the feature for the input sample without the need of retraining, avoiding feature bias toward old classes. Third, to equip the framework with the capability of explicitly separating the unlabeled instances of seen classes from those of unseen ones, we introduce an Open-Set Aware (OSA) classifier in Prompt-SSLC to dynamically route the input samples for classification – more reliable for seen categories and discovery – essential for unseen categories. With all these innovative designs in Prompt-SSLC, it can faithfully handle the category discovery problem in an on-the-fly setting.

We thoroughly evaluate Prompt-SSLC on public benchmarks, encompassing both generic fine-grained datasets, demonstrating superior performance. Particularly for fine-grained datasets, Prompt-SSLC outperforms PHE Zheng et al. (2025) by 10% in new class accuracy, while for coarse-grained tasks, it surpasses SMILE Du et al. (2023) by 20% in new class accuracy. These consistent advances across granularities highlight the adaptability and robustness of our framework.

In summary, we make the following contributions in this paper: **(i)** We propose a unified framework, named Prompt-SSLC, for the emerging open-world challenge of on-the-fly category discovery (OCD). **(ii)** At the core of Prompt-SSLC are our innovative SSLC designs associated with a Distance-Aware prototype distribution update scheme, dual prompt learning to repurpose vision foundation models for OCD, and an Open-Set-Aware classification module to equip the framework with the unknown awareness of the framework. **(iii)** We thoroughly evaluate our methods on public benchmarks across generic and fine-grained datasets, obtaining superior performance, significantly outperforming the previous SOTA methods.

## 2 RELATED WORKS

**Category Discovery**   Category discovery addresses the challenge of transferring knowledge from labeled data to categorize unlabeled data containing instances from known and unknown categories. The task was initially studied as Novel Category Discovery (NCD) (Han et al., 2019). Conventional NCD methods (Han et al., 2021; Zhao & Han, 2021; Jia et al., 2021; Fini et al., 2021) assume unlabeled data contains only novel classes, limiting their practicality. Generalized Category Discovery (GCD) Vaze et al. (2022) relaxes this assumption by allowing unlabeled data to include both known and novel classes. A series of methods have been proposed to address GCD from different perspectives, such as parametric classification (Wen et al., 2023), effective positive sample utilization (Hao et al., 2024), spatial prompt learning (Wang et al., 2024c), attention guidance (Lin et al., 2024), GMM-based representation learning (Zhao et al., 2023), debiased curriculum learning (Liu & Han, 2025), hyperbolic representation learning (Liu et al., 2025). There are also works trying to study the category discovery problem under different settings, such as Continuous Category Discovery (CCD) (Zhang et al., 2022; Cendra et al., 2024), Federated Generalized Category Discovery (Fed-GCD) (Pu et al., 2024), Active Generalized Category Discovery (Ma et al., 2024), Semantic Category Discovery (SCD) (Han et al., 2024), On-the-fly Category Discovery (OCD) (Du et al., 2023) and more. Among other settings, CCD is the most relevant setting to OCD but different in two critical ways. First, CCD takes a collection of unlabeled samples at each time step. Second, CCD needs to optimize the model on the unlabeled samples at each time step. In contrast, OCD is tasked to processes streaming data (regardless of single or multiple instances) without the need of further model training once trained on the labeled data.

**On-the-fly Category Discovery**   OCD is a new and challenging setting of category discovery and the study in this task remains sparse. To the best of our knowledge, there are only two methods, SMILE (Du et al., 2023) and PHE (Zheng et al., 2025), aiming to address this challenge. SMILE Du et al. (2023) employs a hash-based representation for category identification, where the sign of each instance's low-dimensional hash code (*e.g.*, 12-bit) determines its class. A regularization loss is proposed to enforce uniqueness in hash codes for labeled data. However, the short code length inherently limits discriminative power, hindering generalization to unlabeled categories. PHE Zheng et al. (2025) addresses hash sensitivity by mapping classes to prototypical hash centers. Also, a center separation loss is proposed to constrain a minimal separation distance between hash centers. However, its reliance on discrete hash spaces introduces scalability limitations: the finite combinatorial capacity of the hash space restricts adaptability to evolving data streams, making it unsustainable for lifelong learning scenarios. In contrast, we devise a semi-supervised sequential leader clustering framework, which creates new prototypes and dynamically updates prototype parameters regarding the incoming data stream, thereby overcoming the scalability and adaptability shortcomings.

**Open Set Recognition**   Open Set Recognition (OSR) is tasked with rejecting samples from unfamiliar classes w.r.t. the training data at test time. It differs significantly from category discovery in that categorization on the unknown data is not considered. The OSR was initially formalized by Scheirer Scheirer et al. (2012), with OpenMax Bendale & Boult (2015) being the first deep learning method to calibrate classifier confidence scores to distinguish known and unknown classes. Subsequent works introduced diverse strategies including generative methods (Neal et al., 2018; Kong & Ramanan, 2021), reconstruction-based methods (Yoshihashi et al., 2019; Sun et al., 2020), and prototype-based methods (Neal et al., 2018; Kong & Ramanan, 2021). Generative methods (Neal et al., 2018; Kong & Ramanan, 2021) synthesize semantically similar but category-agnostic images to train an open-set classifier using carefully crafted counterexamples. Reconstruction-based methods (Yoshihashi et al., 2019; Sun et al., 2020) use poor test-time reconstruction as an indicator of open-set samples. Prototype-based methods (Neal et al., 2018; Kong & Ramanan, 2021) learn prototypes representing known classes with classification determined by distance metrics then flags outliers as unknowns. As studied in (Vaze et al., 2021; Wang et al., 2024a), the Maximum Softmax

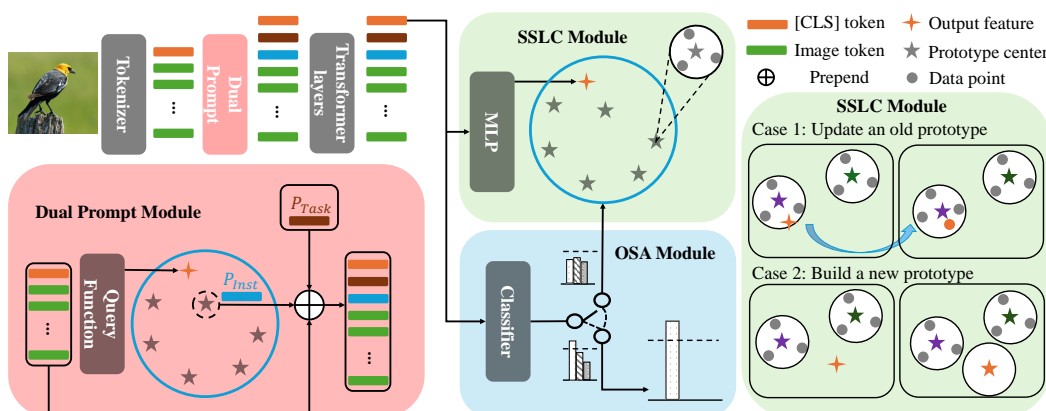

Figure 2: **Overview of the proposed framework.** A Task Prompt ($P_{task}$) and an Instance Prompt ($P_{inst}$) are prepended to image patches as transformer inputs through Dual Prompt Module. The Open-Set-Aware(OSA) classifier serves as a switch to identify and separate confident known-class samples, while uncertain data is routed to the Semi-supervised Sequential Leader Clustering (SSLC) module for on-the-fly category discovery.

Probability (MSP) of a trained classifier serves as a surprisingly strong baseline for OSR. Therefore, we adopt MSP as the OSR scoring rule in our work due to its simplicity and effectiveness. By processing known and open-set samples separately, our approach ensures minimal interference with known-class classification accuracy while continuously discovering novel categories on the fly.

## 3 METHOD

We introduce the OCD framework in Sec. 3.1. Next, we present Prompt-SSLC, designed for robust and efficient OCD. We first propose Semi-supervised Sequential Leader Clustering-Baseline (SSLC-B), a simple yet effective clustering baseline, in Sec. 3.2.2. This baseline is enhanced in Sec. 3.2.3 with a Distance-Aware (DA) update scheme to preserve discriminability between adjacent prototypes, deriving the SSLC algorithm. Building upon this, we integrate a novel dual prompt tuning strategy to enrich feature representations for emerging categories. In Sec. 3.3.1, we reformulate the closed-set classification problem as a GCD task using the *Task Prompt* to encode task-specific information. In Sec. 3.3.2, we further introduce the *Instance Prompt* that captures neighboring prototype relationships. Finally, we design an open-set aware (OSA) classifier in Sec. 3.4 to handle prototype degradation caused by noisy data. The overview of our framework is shown in Fig. 2.

### 3.1 FORMULATION OF OCD

The OCD task leverages labeled data to perform generalized category discovery on streaming unlabeled data without the need of retraining or fine-tuning on the incoming unlabeled data, enabling real-time identification of novel categories alongside classification of unlabeled instances. OCD operates through two distinct phases:

**Training phase** The model is trained on labeled data $D^{\mathrm{l}} = \{(x_i^{\mathrm{l}}, y_i^{\mathrm{l}})\}_{i=1}^{N}$, where $y_i^{\mathrm{l}} \in C_{\mathrm{old}}$ and $C_{\mathrm{old}} = \{1, 2, \ldots, c\}$ represents the set of known (or old) categories.

**Inference phase** The model processes an unlabeled data stream $D^{\mathrm{u}} = \{x_i^{\mathrm{u}}\}_{i=1}^{\infty}$ where the samples belongs to an expanded label space $C = C_{\mathrm{new}} \cup C_{\mathrm{old}}$. At each time step $t$, the cardinality $\hat{K}_t = |\hat{C}_{\mathrm{new}}|$ is dynamically estimated via $\hat{K}_t = g(\{x_i^u\}_{i=1}^{t}, C_{\mathrm{old}})$ where $g(\cdot)$ denotes an adaptive clustering mechanism that incrementally adjusts to incoming data.

This formulation enables on-the-fly generalized category discovery under real-world constraints, where data arrive sequentially and novel classes emerge unpredictably over time. The approach is highly practical, eliminating the need for model tuning during inference, pre-collecting all unlabeled data, or full dataset access while continuously estimating new class counts.

### 3.2 SEMI-SUPERVISED SEQUENTIAL LEADER CLUSTERING (SSLC)

#### 3.2.1 REVISITING SEQUENTIAL LEADER CLUSTERING

The objective of OCD is to predict the label $\hat{y}_i^{\mathrm{u}}$ for each unlabeled instance $x_i^{\mathrm{u}}$. If the predicted label corresponds to an existing class in $C$, *i.e.*, $\hat{y}_i^{\mathrm{u}} \in C$, the instance is classified into that class. If the

predicted label represents a novel class $\hat{y}_i^{\mathrm{u}} \notin C$, category discovery methods, such as semi-supervised $k$-means (Vaze et al., 2022) and pseudo-label refinement techniques (Han et al., 2021; Zhao & Han, 2021; Jia et al., 2021; Fini et al., 2021), prove unsuitable for online scenarios due to their reliance on static datasets and prior knowledge of total category counts.

Sequential Leader Clustering (SLC) appears to be a natural baseline for OCD, due to its sequential clustering nature. It operates through an online prototype update mechanism that dynamically creates clusters based on distance thresholds. For each incoming instance $x_i^{\mathrm{u}}$, SLC calculates distances to all existing prototypes and assigns the instance to the nearest prototype if their distance falls below a predefined radius. When no prototype satisfies this criterion, SLC creates a new cluster centered at $x_i^{\mathrm{u}}$. This approach eliminates the need for prior knowledge of category counts and supports streaming data processing. However, SLC operates purely unsupervised, disregarding available labeled data that could enhance cluster discriminability and alignment with known classes.

To overcome this limitation of the missing knowledge from labeled data, we propose Semi-supervised Sequential Leader Clustering (SSLC), with the benefit of (1) labeled data guides prototype initialization to avoid semantically inconsistent clusters, and (2) Distance-Aware prototype distribution update scheme to maintain separation between adjacent prototypes.

### 3.2.2 SEMI-SUPERVISED SLC FOR OCD

We establish the Semi-supervised Sequential Leader Clustering baseline (SSLC-B) by **(1)** maintaining prototypes for known classes and **(2)** dynamically creating prototypes for new classes from streaming data, as outlined in Algorithm 1. The prototype list is initialized using labeled data, where each known class prototype is computed as the class-wise mean of feature representations. At each time step $t$, the system contains $\hat{K}_t$ prototypes, each defined by a center $\mu$ and a size $n$. For an incoming unlabeled data $x_t^{\mathrm{u}}$, the algorithm identifies the nearest prototype $(\mu_{(1)}, n_{(1)})$ with the $\ell_2$ distance $d_{(1)}$. If $d_{(1)}$ exceeds a pre-defined threshold $\tau$, a new cluster is initialized with $x_t^{\mathrm{u}}$. Otherwise, the

---

**Algorithm 1** SSLC-Baseline (SSCL-B)

---

1: Initialize prototype list $\{\mu_i\}_{i=1}^{|C_{\mathrm{old}}|}$ with labeled data
2: Initialize $\hat{K}_t = |C_{\mathrm{old}}|$
3: **for** $t = 1, 2, \ldots$ **do**
4:     **for** $i = 1, 2, \ldots, \hat{K}_t$ **do**
5:         $d_i = ||x_t^{\mathrm{u}} - \mu_i||_2$
6:     **end for**
7:     Sort $d$ in ascending order to have $d_{(1)}, d_{(2)}, \ldots$
8:     **if** $d_{(1)} \geq \tau$ **then**
9:         $\hat{K}_t \leftarrow \hat{K}_t + 1$
10:        $\mu_{\hat{K}_t} \leftarrow x_t^{\mathrm{u}}$ and $n_{\hat{K}_t} \leftarrow 1$
11:     **else**
12:        $\mu_{(1)} = \mu_{(1)} + (x_t - \mu_{(1)}) * 1/n_{(1)}$
13:        $n_{(1)} = n_{(1)} + 1$
14:     **end if**
15: **end for**

---

data point is assigned to the nearest prototype, and its center parameter and size are updated. Specifically, SSLC-B updates center paraemter with this scheme $\mu_{(1)} = \mu_{(1)} + (x_t - \mu_{(1)}) * 1/n_{(1)}$. By predefining prototypes for known classes through this supervised approach, SSLC-B emerges as a robust baseline that strategically preserves prior knowledge while enabling effective novel class discovery.

### 3.2.3 DISTANCE-AWARE UPDATE SCHEME

The current SSLC-B framework exhibits a critical limitation: unconstrained center adjustments under noisy data distributions induce progressive prototype overlap, as visualized in Fig. 3 (left). This degradation occurs when successive updates disproportionately attract prototypes toward outlier instances, collapsing inter-cluster margins. To address this, we propose to incorporate inter-prototype distance constraints during center updates. This is achieved through two novel update schemes, "push" and "rotation", which explicitly regulate prototype spacing to mitigate overlap while maintaining clustering integrity.

**Push scheme** This scheme simultaneously attracts $\mu_{(1)}$ toward $x_t$ and repels $\mu_{(2)}$ from the observed data point, as formalized in Eq. (1).

$$\begin{aligned} \mu_{(1)} &= \mu_{(1)} + (x_t - \mu_{(1)}) * 1/n_{(1)} \\ \mu_{(2)} &= \mu_{(2)} - (x_t - \mu_{(2)}) * 1/n_{(2)} \end{aligned} \qquad (1)$$

**Rotation scheme** The rotation scheme regulates the distance between $\mu_{(1)}$ and $\mu_{(2)}$ by enforcing constrained movement, which establishes a rotational trajectory about $\mu_{(2)}$. This mechanism guides

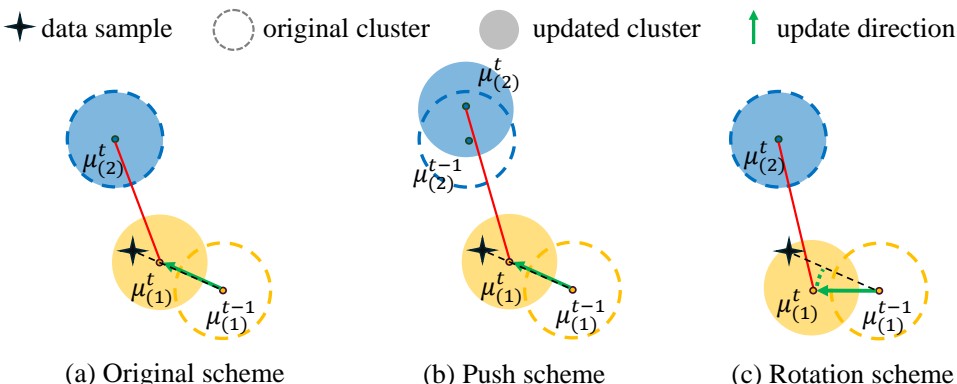

(a) Original scheme          (b) Push scheme          (c) Rotation scheme

Figure 3: **Update scheme comparison.** (a) Original scheme adjusts the nearest prototype. (b) Push scheme simultaneously attracts the nearest prototype and repels the second-nearest. (c) Rotation scheme enhances cluster separation while preserving the second-nearest prototype.

$\mu_{(1)}$ toward $x_t$ while preserving inter-prototype spacing, as formalized in Eq. (2).

$$\mu_{(1)} = \mu_{(1)} + (x_t - \mu_{(1)}) * 1/n_{(1)} + k * (x_t - \mu_{(2)}) \tag{2}$$

Both update schemes produce significantly more distinguishable prototypes than SSLC-B's original mechanism, with the rotation scheme demonstrating superior performance by achieving an ideal balance between prototype adaptation and separation preservation. Consequently, we adopt the **rotation** scheme to establish our SSLC framework.

### 3.3 PROMPT TUNING ADAPATION

#### 3.3.1 *Task Prompt* FOR GCD SIMULATION

Existing OCD methods (Du et al., 2023; Zheng et al., 2025) rely on fully supervised training using labeled data composed solely of known classes, thereby reducing the problem to a closed-set classification task. This approach has a critical shortcoming: the resulting representations become over-specialized for known class discrimination at the expense of foundation models' inherent generalization capabilities, while the closed-set optimization goal misaligns with OCD objective of novel class identification during inference. To address this limitation, we introduce (1) partial label masking that reformulates the training objective to preserve generalization, and (2) a task-aligned prompting strategy that activates the foundation model's inherent generalization capacity while maintaining discriminative power for known classes.

To simulate GCD, partial label masking is employed by dividing the original classes into two mutually exclusive subsets: $C_{pseudo}^{old}$ and $C_{pseudo}^{new}$. The pseudo-old dataset comprises instances from $C_{pseudo}^{old}$, while the pseudo-new dataset contains instances from $C_{pseudo}^{new}$ with their labels masked to reflect unknown categories. This approach enables controlled simulation of novel class discovery within a structured framework. Consequently, supervised contrastive loss (Khosla et al., 2020) is applied to the pseudo-old dataset and self-supervised contrastive loss (Chen et al., 2020) is applied to the pseudo-new dataset.

Directly fine-tuning the model parameters would compromise the foundation model's inherent generalization capabilities. *Task Prompt* address this by acting as lightweight vectors to prioritize task-relevant features, which preserves the foundation model's generalization power while aligning its focus with the OCD objective – balancing known-class discrimination and novel-class sensitivity. To enhance task-specific learning alongside partial label masking, we integrate *Task Prompt* ($P_{task}$) into foundation models. Our framework employs a Vision Transformer (ViT) as its backbone encoder. In this architecture, an input image $X$ is partitioned into $n$ patches $(x_1, \ldots, x_n)$. These patches are projected into a $d$-dimensional latent space via a shared linear layer $e$, combined with a learnable classification token `CLS` with the equation $E^0 = e(x_1, \ldots, x_n)$. The attention mechanism is then expressed as:

$$[\text{CLS}^i, E^i] = L_i([\text{CLS}^{i-1}, E^{i-1}]) \tag{3}$$

where $L_i$ is the $i$-th transformer layer. Inspired by VPT-Deep (Jia et al., 2022), we prepend learnable prompt vectors $P_{task}$ to the input of each transformer layer. The attention mechanism is modified in eq. (4)

$$[\text{CLS}^i, \_, E^i] = L_i([\text{CLS}^{i-1}, P_{task}^{i-1}, E^{i-1}]). \tag{4}$$

### 3.3.2 *Instance Prompt* TO HINT PROTOTYPE DISTRIBUTION

Apart from $P_{task}$ that encodes general task information, we propose *Instance Prompt* ($P_{inst}$) to leverage near-neighbor prototype information and hint at the data distribution.

The training involves two sequential feedforward passes. In the first pass, prompt parameters remain excluded from the input space, and the CLS token from the backbone network, denoted as $z_i = f(x_i)$, serves as the queried feature. The $\ell_2$ distances between this feature and all cluster centers are computed, followed by the identification of the $k$ nearest clusters. The mean parameters of these clusters are concatenated to construct the *Instance Prompt*, that is, $P_{inst} = [\mu_{(1)}, \mu_{(2)}, \ldots, \mu_{(k)}]$. In the second feedforward pass, $P_{inst}$ is integrated into each transformer layer $L_i$, modifying the input space as follows:

$$[\text{CLS}^i, \_, E^i] = L_i([\text{CLS}^{i-1}, P_{task}^{i-1}, P_{inst}^{i-1}, E^{i-1}]). \tag{5}$$

This system offers a unique advantage by dynamically extending prototype pools during inference without requiring tuning. This capability stems from the integration of $P_{inst}$ with SSLC, where the expandable prototype list used in SSLC is shared with the prompt pool. As a result, the combined framework generates more discriminative features and achieves tighter cluster formations compared to non-prompt-based models.

### 3.4 SAMPLE ROUTING WITH OPEN-SET AWARE CLASSIFIER

To leverage open-set recognition, we introduce an Open-set Aware (OSA) classifier designed to filter low-confidence samples. This classifier treats all novel classes as a single superclass, enabling a unified classification task across $c + 1$ classes: $c$ known classes and one aggregated novel superclass. Inspired by (Vaze et al., 2021; Wang et al., 2024a), we adopt the Maximum Softmax Probability (MSP) as the OSA scoring rule, defined as $S(y \in C_{old}|x) = max_{y \in C_{old}} P(y|x)$ where $P(\cdot)$ is the softmax output vector. Unlabeled samples with sufficiently high OSA scores, indicating high confidence in belonging to a known class, are directly assigned to the corresponding old class with the highest softmax score. In contrast, low-confidence samples are routed to the SSLC module for further analysis. This approach ensures that the integrity of established prototypes is preserved while efficiently isolating novel class instances for the subsequent processing.

## 4 EXPERIMENTS

### 4.1 EXPERIMENT SETUP

#### 4.1.1 DATASETS

Table 1: **Statistics of datasets.** Number of classes $C$ and number of samples $D$

|  | CUB | Scars | Herb | CIFAR10 | CIFAR100 | ImageNet100 |
|---|---|---|---|---|---|---|
| $|C_{old}|$ | 100 | 98 | 341 | 5 | 80 | 50 |
| $|C|$ | 200 | 196 | 683 | 10 | 100 | 50 |
| $|D^l|$ | 1.5k | 2.0k | 8.9k | 12.5k | 20.0k | 31.9k |
| $|D^u|$ | 4.5k | 6.1k | 25.4k | 37.5k | 30.0k | 95.3k |

We conduct experiments on three generic image classification datasets: CIFAR10 (Krizhevsky et al., 2009), CIFAR100 (Krizhevsky et al., 2009), and ImageNet100 (Russakovsky et al., 2015), alongside three fine-grained image datasets: CUB (Wah et al., 2011), Scars (Krause et al., 2013), and Herbarium19 (Tan et al., 2019). The statistics of those datasets are presented in Tab. 1. We follow the data split of previous works (Vaze et al., 2022; Du et al., 2023). $C_{old}$ classes are sampled to build the labeled set, then about half of images from the labeled set constitute the labeled dataset $D^l$. The remaining images, which come from all classes, constitute the unlabeled dataset $D^u$.

#### 4.1.2 EVALUATION PROTOCOL

For each dataset, the model is trained on $D^l$, while $D^u$ is sequentially shuffled and presented during inference. We propose a modified Hungarian matching strategy for evaluation. We leverage the labeled dataset $D^l$ to create the prototypes of each old class with its corresponding feature representation. For old categories, the old-class accuracy represents the percentage of samples from the unlabeled dataset $D^u$ that correctly match their respective old-class prototypes. For novel categories, we compute the new-class accuracy by applying Hungarian matching to align model predictions with ground-truth labels. This framework enables separate performance evaluations for old and new classes, effectively reducing bias from previously learned class representations.

Table 2: **Comparison with state-of-the-art methods on coarse-grained datasets.**

| Methods | CIFAR100 | | | ImageNet100 | | | CIFAR10 | | |
| --- | --- | --- | --- | --- | --- | --- | --- | --- | --- |
| | All | Old | New | All | Old | New | All | Old | New |
| SLC Hartigan (1973) | 59.88 | 73.42 | 34.51 | 51.26 | 85.32 | 34.23 | 36.52 | 82.45 | 13.58 |
| SMILE Du et al. (2023) | 59.49 | 70.31 | 37.85 | 51.33 | 83.12 | 35.44 | 36.65 | 74.57 | 17.68 |
| SSLC-B | 66.58 | 76.58 | 46.52 | 63.60 | 88.35 | 51.23 | 46.76 | 86.56 | 26.88 |
| SSLC | 72.21 | 82.36 | 51.92 | 66.53 | 88.65 | 55.47 | 49.57 | 87.53 | 30.57 |
| Prompt-SSLC | **76.04** | **87.31** | **53.51** | **70.35** | **91.13** | **60.02** | **53.34** | **90.98** | **34.52** |

Table 3: **Comparison with state-of-the-art methods on fine-grained datasets.**

| Methods | CUB | | | Scars | | | Herb | | |
| --- | --- | --- | --- | --- | --- | --- | --- | --- | --- |
| | All | Old | New | All | Old | New | All | Old | New |
| SLC Hartigan (1973) | 39.44 | 78.32 | 20.01 | 36.75 | 71.54 | 20.57 | 32.89 | 56.05 | 20.41 |
| SMILE Du et al. (2023) | 35.23 | 60.23 | 22.73 | 31.79 | 55.47 | 19.95 | 26.27 | 45.06 | 16.87 |
| PHE Zheng et al. (2025) | 39.38 | 69.14 | 24.51 | 37.23 | 65.38 | 23.14 | 31.32 | 52.33 | 20.81 |
| SSLC-B | 50.41 | 88.28 | 31.48 | 48.96 | 85.31 | 30.82 | 34.68 | 58.65 | 21.78 |
| SSLC | 50.87 | 88.71 | 31.93 | 50.22 | 85.58 | 32.54 | 38.31 | 63.20 | 24.91 |
| Prompt-SSLC | **52.37** | **90.10** | **33.51** | **50.94** | **86.50** | **33.16** | **44.96** | **72.19** | **30.13** |

### 4.1.3 IMPLEMENTATION DETAILS

The model architecture comprises four key components: a dual prompt, a backbone encoder, an MLP projection head, and a classification head. The backbone encoder employs a ViT-B initialized from DINOv2 (Oquab et al., 2023), where only the final transformer block is fine-tuned while the remaining layers remain frozen. $P_{task}$ and $P_{inst}$ are inserted between the `CLS` token and image patch tokens. Both the MLP projection head and the single-layer classification head process the `CLS` token's feature output. The MLP head produces a 4096-dimensional $L2$-normalized embedding for the SSLC module, while the classification head generates logits corresponding to the number of old classes for the OSA classifier. To ensure a fair comparison, we employ the same DINOv2 backbone as the feature extractor for reproducing the experimental setups of SMILE (Du et al., 2023) and PHE (Zheng et al., 2025). We configure the MLP output dimension to 12 for SMILE and 64 for PHE, adhering to their respective official implementations. Training configurations vary by dataset granularity: fine-grained datasets undergo 100 epochs, whereas coarse-grained datasets use 50 epochs. Optimization employs the Adam optimizer ($\beta_1$=0.9, $\beta_2$=0.999) with a 1e-4 initial learning rate and a cosine learning rate schedule in a batch size of 128. The $P_{task}$ length is fixed at 3, while top-$k$ value for $P_{inst}$ is 3.

### 4.2 COMPARISON WITH SOTA METHODS

Our method is evaluated on six benchmark datasets against state-of-the-art competitors including SLC (Hartigan, 1973), SMILE (Du et al., 2023), and PHE (Zheng et al., 2025). Results for coarse-grained and fine-grained tasks are presented in Tab. 2 and Tab. 3. SSLC-B achieves significant improvements across all benchmarks over previous SOTAs, validating the efficacy of prebuilt known-class prototypes in discriminative category discovery. In addition, SSLC (Sec. 3.2.3) outperforms SSLC-B (Sec. 3.2.2) on all metrics, with an average 3% increase in new class accuracy, indicating the effectiveness of the DA prototype update scheme. For fine-grained datasets, Prompt-SSLC outperforms PHE (Zheng et al., 2025) by 10% in new class accuracy and exceeds SSLC-B by 4%, demonstrating superior generalization to unseen fine-grained categories. For coarse-grained tasks, Prompt-SSLC surpasses SMILE (Du et al., 2023) by 20% and SSLC-B by 9% in new class accuracy. These consistent advances across granularities highlight the adaptability and robustness of our framework.

### 4.3 ABLATION STUDY

In this section, we conduct ablation studies to examine the effectiveness of individual components. Compared to SSLC-B, our approach has four key innovations: a DA update scheme, a $P_{task}$ and a $P_{inst}$, and an OSA classifier. The effectiveness of these components and design choices is systematically validated in Tab. 4. Further studies on effects of hyperparameters and robustness of Prompt-SSLC under varying ratio of labeled to unlabeled classes are shown in Appendix A.

**Designs for DA scheme.** We evaluate two implementations of the DA update scheme—the Push scheme and the Rotation scheme—alongside their combination, integrated with SSLC-B, as shown in the top block of Tab. 4. The Rotation scheme demonstrates superior performance, improving new

Table 4: **Ablation study on the components of our approach.**

| | Methods | Scars | | | IM100 | | |
|---|---|---|---|---|---|---|---|
| | | All | Old | New | All | Old | New |
| Reference | SLC | 36.75 | 71.54 | 20.57 | 51.26 | 85.32 | 34.23 |
| | SSLC-B *w.* Original scheme | 48.96 | 85.31 | 30.82 | 63.60 | 88.35 | 51.23 |
| | SSLC-B *w.* Push scheme | 49.39 | 85.42 | 31.36 | 65.52 | 88.48 | 54.03 |
| | SSLC-B *w.* Rotation scheme (SSLC) | **50.22** | **85.58** | **32.54** | **66.53** | **88.65** | **55.47** |
| | SSLC-B *w.* Both scheme (SSLC) | 49.14 | 85.36 | 31.03 | 64.36 | 88.41 | 52.38 |
| Reference | Prompt-SSLC (Ours) | **50.94** | **86.50** | **33.16** | **70.35** | **91.13** | **60.02** |
| | *w.o.* Rotation scheme | 49.81 | 86.45 | 31.48 | 67.10 | 90.54 | 55.41 |
| | *w.o.* OSA classifier | 50.36 | 85.76 | 32.66 | 66.78 | 88.94 | 55.70 |
| | *w.o. Task Prompt* | 50.61 | 85.81 | 33.02 | 68.48 | 89.96 | 57.73 |
| | *w.o. Instance Prompt* | 50.46 | 85.89 | 32.75 | 67.71 | 90.21 | 56.44 |

Table 5: **Comparison of task prompt with PEFTs.**

| PEFT method | Scars | | | CIFAR100 | | |
|---|---|---|---|---|---|---|
| | All | Old | New | All | Old | New |
| Task Prompt | 50.94 | 86.50 | 33.16 | 76.04 | 87.31 | 53.51 |
| Lora (Hu et al., 2022) | 49.70 | 86.35 | 31.37 | 75.10 | 86.65 | 51.97 |
| Adaptor (Houlsby et al., 2019) | 49.22 | 85.86 | 30.93 | 74.35 | 86.13 | 50.85 |

class accuracy over the Push variant by 1%. The Push scheme risks destabilizing the second nearest prototype due to noisy inputs, whereas the Rotation scheme ensures a more stable DA prototype distribution update. The combination of both schemes underperforms the Rotation scheme alone, as the Push scheme disrupts the Rotation scheme's design goal of enhancing inter-prototype separation while preserving neighboring prototype stability.

**Effectiveness of different components.** The bottom block of Tab. 4 evaluates the contributions. Removing the OSA classifier causes a significant decline in old class accuracy by 2%, as it filters low-confidence data to prevent prototype perturbation by noisy samples. Ablating $P_{task}$ reduces performance in all accuracy by 1%, and removing $P_{inst}$ decreases new class accuracy by 2%, underscoring the necessity of the dual prompt tuning.

**Task prompt design.** We conduct a study comparing our task prompt approach with parameter-efficient fine-tuning (PEFT) methods, including Adapter (Houlsby et al., 2019) (r=8) and Lora (Hu et al., 2022) (r=4). Results presented in Tab. 5 show that task prompt tokens outperform Adapter and LoRA by approximately 2% on the Stanford Cars (Krause et al., 2013) and CIFAR-100 (Krizhevsky et al., 2009) datasets. This performance gain stems from the task prompt's ability to encode category discovery-specific information, rather than merely adding parameters. The task prompt tokens are designed to capture general task information relevant to category discovery, guiding the model to extract features tailored for clustering unseen data.

## 5 CONCLUSION

This work introduces a novel framework Prompt-SSLC for OCD. The proposed framework systematically combines three key components to overcome limitations in discrete hashing-based approaches and SLC. First, SSLC guides prototype initialization with labeled data to avoid semantically inconsistency and employs a Distance-Aware prototype distribution update scheme to dynamically maintain seperation between adjacent prototypes. Next, we enhance feature representation for novel classes by introducing dual prompt tuning: a Task Prompt and an Instance Prompt. These components mitigate old-class feature bias while enabling dynamic representation updates without fine-tuning the foundation model. In addition, an Open-Set-Aware classifier is utilized to identify and filter ambiguous samples through uncertainty estimation, thereby preventing prototype contamination during incremental update. Together, these components systematically address OCD. Extensive experiments on generic and fine-grained benchmarks demonstrate that our method outperforming existing approaches by wide margins, establishing the state-of-the-art for OCD.

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

# Prompt-SSLC: Prompting Semi-supervised Sequential Leader Clustering for On-the-fly Category Discovery

## — Supplementary Material —

## A  ADDITIONAL EXPERIMENTS AND ANALYSIS

**Comparison with GCD methods.**  We conduct comparative experiments with GCD (Vaze et al., 2022) and SimGCD (Wen et al., 2023) on CUB and ImageNet100 datasets. To ensure fair comparisons, all methods utilize the estimated number of categories as proposed in GCD (Vaze et al., 2022) and adopt the DINO backbone architecture. Two variants of Prompt-SSLC are evaluated: the first variant is trained exclusively on labeled data $D^l$, aligning with the OCD setting, while the second variant see both $D^l$ and $D^u$ during training to assess GCD performance. The results (as shown in Tab. I) demonstrate the effectiveness of our Prompt-SSLC. Prompt-SSLC learnt on $D^l$ achieves a 10% improvement in old-class accuracy over GCD and SimGCD, demonstrating robust classification capabilities. With access to both labeled and unlabeled training data, the new-class accuracy increases by 5% compared to the labeled-only variant, resulting in an overall accuracy that surpasses GCD by approximately 2%. This shows that Prompt-SSLC can achieve a similar performance as of GCD without the need of collecting all unlabelled data to perform clustering.

Table I: **Comparison with GCD methods.**

| Methods | CUB | | | IM100 | | |
|---|---|---|---|---|---|---|
| | All | Old | New | All | Old | New |
| GCD | 47.1 | 55.1 | 44.8 | 72.7 | 91.8 | 63.8 |
| SimGCD | 61.0 | 66.0 | 58.6 | 81.1 | 90.9 | 76.1 |
| Prompt-SSLC (labelled only) | 43.93 | 73.86 | 28.94 | 69.8 | 93.45 | 56.31 |
| Prompt-SSLC | 50.22 | 74.8 | 38.84 | 73.31 | 94.33 | 62.74 |

Table II: **Comparison with GCD methods on testing unlabelled data.**

| Methods | CUB | | | IM100 | | |
|---|---|---|---|---|---|---|
| | All | Old | New | All | Old | New |
| GCD | 52.48 | 56.21 | 46.7 | 78.14 | 91.85 | 64.5 |
| SimGCD | 64.3 | 66.73 | 59.31 | 84.26 | 91.23 | 77.41 |
| Prompt-SSLC (labelled only) | 50.83 | 73.89 | 30.04 | 75.72 | 93.75 | 59.25 |
| Prompt-SSLC | 57.24 | 75.15 | 41.37 | 79.10 | 94.65 | 63.86 |

**Results with variant DINO backbones.**  We perform the main experiments using the DINO backbone architecture and employ the Hungarian matching algorithm as implemented in GCD (Vaze et al., 2022) and SMILE (Du et al., 2023), distinct from the constrained variant described in our main paper. Our method is evaluated on six benchmark datasets against state-of-the-art competitors including SLC (Hartigan, 1973), SMILE (Du et al., 2023), and PHE (Zheng et al., 2025). Results for coarse-grained and fine-grained tasks are presented here. SSLC (Sec. 3.2.3) outperforms SMILE (Du et al., 2023) and PHE (Zheng et al., 2025) across all metrics, with an average 7% increase in new class accuracy, indicating the effectiveness of DA prototype update scheme. For fine-grained datasets, Prompt-SSLC outperforms PHE (Zheng et al., 2025) by 10% in new class accuracy and exceeds SSLC by 4%, demonstrating superior generalization to unseen fine-grained categories. For coarse-grained tasks, Prompt-SSLC surpasses SMILE (Du et al., 2023) by 15% in all class accuracy. These consistent advances across granularities highlight the adaptability and robustness of our framework.

Table III: **Comparison with state-of-the-art methods on coarse-grained datasets.**

| Methods | CIFAR100 | | | ImageNet100 | | | CIFAR10 | | |
|---|---|---|---|---|---|---|---|---|---|
| | All | Old | New | All | Old | New | All | Old | New |
| SLC Hartigan (1973) | 44.36 | 58.98 | 15.10 | 32.92 | 86.55 | 5.22 | 41.54 | 58.29 | 33.29 |
| SMILE Du et al. (2023) | 51.59 | 61.55 | 31.69 | 33.78 | 74.22 | 13.45 | 49.86 | 39.86 | 54.86 |
| PHE Zheng et al. (2025) | 58.13 | 68.67 | 36.85 | 52.75 | 85.61 | 37.58 | 35.97 | 73.12 | 17.34 |
| SSLC | 63.21 | 79.8 | 30.02 | 51.32 | 89.21 | 31.38 | 65.13 | 60.89 | 68.46 |
| Prompt-SSLC | **65.39** | **82.34** | **32.14** | **53.86** | **90.28** | **35.67** | **68.23** | **62.46** | **70.26** |

**Effects of hyperparameters.**  We ablate two hyperparameters: the instance prompt length and the OSA classifier threshold, as presented in Tab. V and Tab. VI. The default configuration uses

Table IV: **Comparison with state-of-the-art methods on fine-grained datasets.**

| Methods | CUB | | | Scars | | | Herb | | |
|---|---|---|---|---|---|---|---|---|---|
| | All | Old | New | All | Old | New | All | Old | New |
| SLC Hartigan (1973) | 31.3 | 48.5 | 22.7 | 24.0 | 45.8 | 13.6 | 14.92 | 27.44 | 8.14 |
| SMILE Du et al. (2023) | 32.2 | 55.8 | 22.9 | 26.2 | 46.7 | 16.3 | 22.9 | 39.29 | 14.09 |
| PHE Zheng et al. (2025) | 36.4 | 55.8 | 27.0 | 31.3 | 61.9 | 16.8 | 25.5 | 44.3 | 16.03 |
| SSLC | 47.09 | 72.80 | 34.23 | 39.48 | 76.77 | 21.69 | 28.45 | 52.32 | 16.87 |
| Prompt-SSLC | **49.43** | **74.57** | **36.88** | **45.18** | **79.23** | **28.16** | **31.21** | **56.74** | **19.47** |

Table V: **Sensitivity analysis of instance prompt**

| Instance prompt length | Scars | | | CIFAR100 | | |
|---|---|---|---|---|---|---|
| | All | Old | New | All | Old | New |
| 1 | 50.63 | 86.03 | 32.93 | 75.63 | 86.96 | 53.15 |
| 3 | 50.94 | 86.50 | 33.16 | 76.04 | 87.31 | 53.51 |
| 5 | 50.91 | 86.65 | 33.06 | 76.13 | 87.56 | 53.24 |

three instance prompts, balancing computational efficiency and representation adaptability for online clustering. We evaluated prompt lengths (1, 3, 5) on CIFAR-100 and SCARS datasets. Fewer prompts limit dynamic representation adaptability, while more prompts increase computational overhead without significant performance gains. The OSA classifier threshold is set at the 95th percentile of Euclidean distances between labeled data samples and their prototype centers in the training set, computed during initialization. This data-driven approach ensures the threshold aligns with each dataset's distribution. Sensitivity analysis on CIFAR-100 and SCARS datasets, testing the 90th, 95th, and 99th percentiles, reveals that higher percentiles improve known class accuracy but reduce new class accuracy. The 95th percentile optimally balances known class classification and new class discovery.

**Varying ratios of labeled to unlabeled classes.** By default, the number of labeled classes is set to half the total number of classes. To evaluate the robustness of our Prompt-SSLC framework under different labeled-to-unlabeled class ratios, we conduct experiments on the Stanford Cars (Krause et al., 2013) and ImageNet-100 datasets. We test configurations including 10% labeled to 90% unlabeled, 30% labeled to 70% unlabeled, and the baseline 50% labeled to 50% unlabeled settings, presented in Tab. VII These experiments validate that Prompt-SSLC maintains robust performance across diverse labeled data proportions.

**Prototype drift.** Prototype drift is a known challenge in SLC and other clustering methods like $K$-Means, where older members may no longer align with updated cluster prototypes. To empirically validate the effectiveness of our DA update scheme in preventing drift, we conducted an experiment comparing cluster assignment consistency between SSLC and the original SLC. We measured the percentage of samples receiving different cluster assignments between the midpoint and the end of a streaming data sequence. The results show that SSLC reassigns only approximately 8% of samples, compared to over 14% for original SLC. This demonstrates that our DA update scheme significantly mitigates prototype drift, ensuring more stable and meaningful cluster assignments over time.

**Parameter count.** Tunable parameters of Prompt-SSLC mainly consists of backbone parameters and prompt parameters. THe amount of backbone parameters is about 7.09 Million, which is the same as previuous works (Du et al., 2023; Zheng et al., 2025). The extra prompt parameters (including both task prompt and instance prompt) accounts for 55296, which is only 0.71% of the backbone parameter. With a few addition of prompt parameters, Prompt-SSLC achieves significant gains on all metrics compared to previous methods, demonstrating the efficiency and effectiveness of prompt module.

## B FURTHER VISUALIZATION

**DA update scheme prevents overlap.** We analyze the inter-prototype distance distribution with and without the Distance-Aware (DA) update scheme by computing the $\ell_2$ distance between each prototype and its nearest neighboring prototype on StanfordCar (Krause et al., 2013), as shown in Fig. I. The original SLC update scheme exhibits smaller inter-prototype distances, indicating prototype overlap and reduced discriminability. The rotation scheme effectively mitigates this issue, increasing inter-prototype distances to establish clearer decision boundaries.

Table VI: **Sensitivity analysis of threshold value**

| OSA classifier threshold | Scars | | | CIFAR100 | | |
|---|---|---|---|---|---|---|
| | All | Old | New | All | Old | New |
| 90 | 50.80 | 84.51 | 34.02 | 75.14 | 85.58 | 54.26 |
| 95 | 50.94 | 86.50 | 33.16 | 76.04 | 87.31 | 53.51 |
| 99 | 49.98 | 88.38 | 30.76 | 76.32 | 88.25 | 52.46 |

Table VII: **Varying ratios of labeled to unlabeled classes.**

| Ratio (old / new) | Scars | | | IM100 | | |
|---|---|---|---|---|---|---|
| | All | Old | New | All | Old | New |
| 50:50 | 50.94 | 86.50 | 33.16 | 70.35 | 91.13 | 60.02 |
| 30:70 | 43.54 | 75.84 | 27.42 | 63.81 | 83.42 | 54.03 |
| 10:90 | 35.02 | 64.37 | 20.38 | 56.08 | 74.58 | 46.85 |

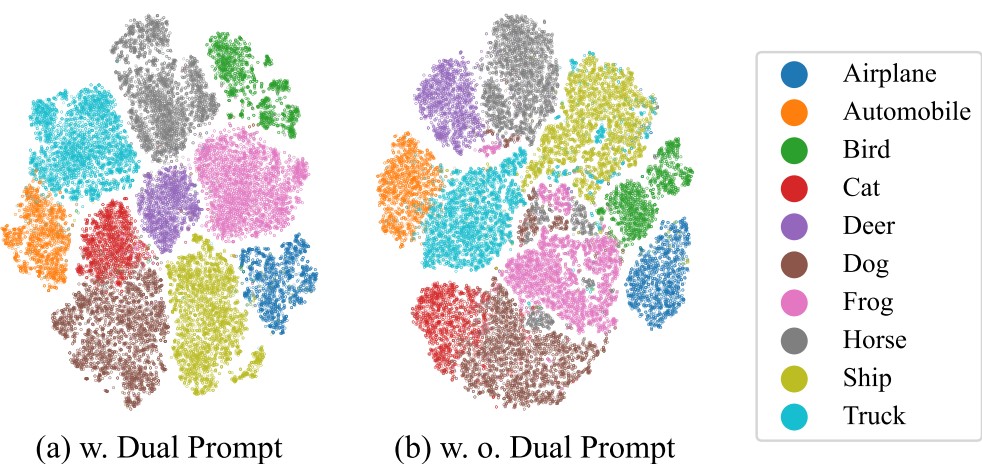

Airplane
Automobile
Bird
Cat
Deer
Dog
Frog
Horse
Ship
Truck

(a) w. Dual Prompt     (b) w. o. Dual Prompt

Figure II: **TSNE visualization of CIFAR10 samples.** (a) is the model with Dual Prompt and (b) is the model without Dual Prompt. The model without Dual Prompt shows confusion between old class features and new class features.

**Dual prompt enhances representations.** We analyze feature representations on CIFAR-10 (Krizhevsky et al., 2009) through $t$-SNE in Fig. II, comparing models with and without the dual-prompt tuning. The baseline lacking dual prompts exhibits degraded feature separation, causing significant overlap between seen and unseen classes under OCD. In contrast, the proposed dual-prompt mechanism yields more discriminative representations with tighter class clusters.

## C   EVALUATION

FRAMEWORK OF MAIN EXPERIMENTS

In this section, we formalize the constrained Hungarian matching that address feature bias toward old classes during evaluation. The core idea is to use $D^l$ to find mapping between features and old categories and to utilize this mapping to guide Hungarian matching. (*e.g.*, in SMILE (Du et al., 2023) case, each old class is uniquely represented by a hash code feature.). We denote $f_i$ for $i = 1, 2, \ldots, c$ to be unique features corresponding to each old class. For all features

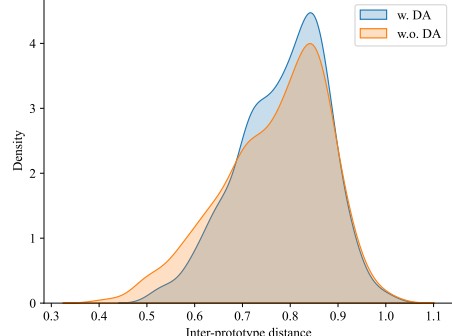

Figure I: **Inter-prototype distance distribution.** SSLC with DA update scheme achieves greater inter-prototype distances, establishing clearer decision boundaries.

extracted from $D^{\mathrm{u}}$, they are compared against all $f_i$ to check if they belong to old classes. This feature is assigned to the old class if this mapping exists. Otherwise, remaining features perform a Hungarian matching with $C_{\mathrm{new}}$. The old-class accuracy is defined as the percentage of old-class samples in $D^{\mathrm{u}}$ that match correctly to corresponding features relative to the total old-class samples. For novel categories, Hungarian matching is exclusively applied to align model predictions with ground-truth labels, enabling new class accuracy computation. Previous literature (Vaze et al., 2022; Du et al., 2023) computes accuracy by matching all extracted features across $C = C_{\mathrm{new}} \cup C_{\mathrm{old}}$. Importantly, this allows the well-built old clusters to be mapped to new categories. We suggest that this makes the model performance biased and does not quite reflect the true OCD setting.

## D  BOARDER IMPACT

This study advances the development of AI systems capable of transitioning from closed-world to open-world settings under changing environments, enabling automated categorization and organization of continuous, real-world data. By improving the robustness of AI models in handling online scenarios, our approach has the potential to enhance applications in domains such as autonomous systems, information retrieval, and large-scale data analysis, fostering more adaptive and scalable AI technologies. However, deploying our method in real-world applications requires careful consideration. The complexity and variability of real-world data, which often exhibit long-tailed distributions, pose significant challenges for model generalization and continual adaptation. Rigorous validation and domain-specific fine-tuning are essential to ensure reliable performance. Additionally, potential risks, such as biased predictions due to imbalanced data or misclassification in critical applications, must be mitigated through transparent evaluation and robust safeguards. These limitations underscore the need for cautious deployment and ongoing research to address open-world challenges effectively.

