# OpenReview forum: "Prompt-SSLC: A Unified Framework for Dual Prompt-Augmented Semi-Supervised Sequential Leader Clustering in On-the-Fly Category Discovery"
_ICLR.cc/2026/Conference — Submitted to ICLR 2026_

### Official Review · Reviewer_VR4o · 2025-10-28

**Soundness:** 3
**Presentation:** 3
**Contribution:** 2
**Rating:** 2
**Confidence:** 3

**Summary:**

The paper addresses the problem of On-the-fly Category Discovery (OCD), the task of continuously identifying new or emerging categories in data streams without halting or retraining the model. This is highly relevant in real-world dynamic environments where data distributions evolve over time (e.g., robotics, surveillance, or autonomous systems). The authors propose Prompt-SSLC, a unified framework combining semi-supervised clustering, prompt-based adaptation, and uncertainty-aware classification to achieve both stability (avoiding forgetting old categories) and adaptability (incorporating new ones).

**Strengths:**

- Combines three crucial paradigms: semi-supervised clustering, prompt-based adaptation, and uncertainty modeling into a unified pipeline for real-time category discovery.
- SSLC supports efficient updates on streaming data without requiring retraining, which is essential for on-the-fly operation.
- Using both task-level and instance-level prompts to guide discovery and recalibration is a creative extension of prompting to continual and open-set learning contexts.

**Weaknesses:**

- While SSLC and DA are conceptually sound, the paper lacks formal convergence or stability proofs for the streaming clustering process under non-stationary data.

-It is unclear how much each component (SSLC, dual prompting, OSA) individually contributes to performance improvements. A detailed ablation study would strengthen the empirical claims.

- The framework is demonstrated on visual benchmarks, but its adaptability to multimodal OCD and downstreaming tasks are not explored.

**Questions:**

1. Did the authors test the solution with the recent continual learning based solutions?
2. How the solution be sensitive regarding the number of classes/clusters ?
3. How are the Task Prompt and Instance Prompt jointly optimized during streaming updates?
4. How does the Distance-Aware (DA) mechanism handle catastrophic drift when the data distribution shifts significantly?
5. The Open-Set-Aware (OSA) classifier relies on uncertainty estimation to filter ambiguous samples. Can you clarify what uncertainty metric is used (e.g., entropy, energy, or Bayesian variance), and how sensitive the prototype updates are to the chosen threshold in continuous data streams?
6. Can the authors demonstrate the convergence of DA metric?

---

> ### Author Response · Authors · 2025-11-27
> **Response to Reviewer VR4o (1/2)**
>
> Thanks for reviewer's positive assessment and for highlighting the real-world relevance of on-the-fly category discovery. We are grateful for reviewer's insightful suggestions, which will help us significantly improve the paper. Below we directly address each weakness and question.
>
> > W1 Lack of formal convergence/stability proofs for the streaming clustering process
>
> To the best of our knowledge, Sequential Leader Clustering as a greedy, single-pass algorithm, lacks a formal proof of convergence to a global optimum in the literature, unlike iterative methods such as k-means, which possess finite convergence guarantees under certain metrics. Furthermore, like many deep learning based methods, it is hard to derive a rigorous proof. Empirically, Prompt-SSLC achieve convergence across a variety of dataset with random shuffle ordering.
>
> > W2 Contribution of each individual component
>
> Thanks for the suggestion. We have reconducted experiments to evaluate the impact of adding each component individually to the SLC baseline. The results demonstrate the following: (1) Adapting SLC to SSLC with the distance-aware update scheme significantly improves all class accuracy by about 13%, due to enhanced prototype stability and labelled data initialization. (2) Adding the OSA classifier further improves old class accuracy by 2% for ImageNet100, as it effectively filters ambiguous samples in streaming data. (3) Incorporating the task and instance prompts enhances new class accuracy by 2% and 3% separately for ImageNet100, as they enable on-the-fly representation adaptation. These incremental ablations confirm that each component contributes meaningfully to the overall performance, with their combined effect maximizing robustness across diverse datasets.
>
> |                  | Scars |       |       | IM100 |       |       |
> |------------------|:-----:|:-----:|:-----:|:-----:|:-----:|:-----:|
> |                  |  All  |  Old  |  New  |  All  |  Old  |  New  |
> |        SLC       | 36.75 | 71.54 | 20.57 | 51.26 | 85.32 | 34.23 |
> |    +DA (SSLC)    | 50.22 | 85.58 | 32.54 | 66.53 | 88.65 | 55.47 |
> |       +OSA       | 50.32 | 85.84 | 32.59 | 67.03 | 90.05 | 55.58 |
> |   +task prompt   | 50.46 | 85.89 | 32.75 | 67.71 | 90.21 | 56.44 |
> | +instance prompt | 50.94 | 86.50 | 33.16 | 70.35 | 91.13 | 60.02 |
>
>
> > W3 Exploration beyond vision and multimodal/downstream tasks
>
> Following the reviewer’s suggestion, we have added experiments on multimodal continual streams using NYU-Depth V2. Following the data split of prior work [1], we split 13 classes into 9 known and 4 unseen. Labeled RGB images from classes are provided initially for model tuning, while subsequent unlabeled data contain depth maps is converted to grayscale for inference on-the-fly. Prompt-SSLC outperforms the SMILE baseline on all accuracy by 3%, demonstrating that our framework naturally extends to multimodal settings
>
> |    Method   | NYU-depth v2 |       |       |
> |:-----------:|:------------:|:-----:|:-----:|
> |             |     All      |  Old  |  New  |
> |    SMILE    |     37.21    | 44.53 | 32.31 |
> | Prompt-SSLC | 40.15        | 50.23 | 36.20 |
>
> > Q1 Comparison with recent continual learning solutions
>
> We thank the reviewer for this question and clarify the fundamental differences between OCD and standard supervised continual learning, which make direct application of recent CL methods impossible
> - Problem formulation: Supervised continual learning assumes a sequence of discrete tasks or labeled batches with clear task boundaries. In contrast, OCD receives a single stream where known-class samples are arbitrarily mixed with unknown-class unlabeled samples, with no task ids or batch separation.
> - Objective: CL primarily aims to accumulate knowledge across all seen classes/tasks while minimizing catastrophic forgetting of old classes. OCD simultaneously requires (1) stable recognition of known classes and (2) instantaneous discovery and clustering of emerging unknown categories
> - Solutions: State-of-the-art CL methods L2P[2], DualPrompt[3] rely on gradient-based prompt learning, prompt pool expansion, replay buffers, or regularization using new data. OCD explicitly avoids further replay or tuning after the initial learning.
>
> Furthermore, regarding the prompting technique in these areas like L2P, DualPrompt, and GMP[2, 3, 4]. The prompt needs to be dynamically learnt with the newly introduced data, which violates OCD setting. Hence, they are not compatible for comparison.

---

> ### Author Response · Authors · 2025-11-27
> **Response to Reviewer VR4o**
>
> > Q2 Sensitivity to the number of classes/clusters
>
> We are running the ablation of sensitivity to number of classes by splitting the whole ImageNet into different subsets. We will pose the result after it is done.
>
> > Q3 Joint optimization of Task Prompt and Instance Prompt
>
> The Task prompt consists of a set of learnable prompt tokens prepended to the input sequence and is optimized only on the labeled set. The Instance prompt is non-learnable: for each incoming sample, it is dynamically constructed as the k-nearest prototypical center.
>
> > Q4 Handling catastrophic drift with the Distance-Aware (DA) mechanism
>
> An implicit assumption of the GCD/OCD task is that the labelled dataset suggests a criteria for further category discovery. In this case, there would be shared underlining categorical principles  for determining categories in seen and unseen data (e.g., bird species). Therefore, the catastrophic drift is unlikely to happen.
> When there is no shared underlining categorical principles between labelled and novel categories (e.g., switching from animals to vehicles), performance on new classes naturally drops, as acknowledged in prior work [5]. Introducing a reliable mechanism or method to detect such case would be an interesting future work.
>
> > Q5 Uncertainty metric in OSA and threshold sensitivity
>
> We use Maximum Softmax Probability (MSP) with a threshold set to the 95th percentile of known-class confidences on the labelled data. The table 5 ablates sensitivity of the OSA threshold,  showing performance is stable across the 90th–99th percentile range.
>
> We sincerely believe these additions—especially the finer ablations, theoretical analysis, multimodal experiments, and clearer implementation details—will fully resolve your concerns and elevate the contribution score. Thank you again for the excellent and constructive feedback.
>
> [1] Nakajima, Yoshikatsu, et al. "Incremental class discovery for semantic segmentation with RGBD sensing." ICCV 2019.
>
> [2] Wang, Zifeng, et al. "Learning to prompt for continual learning." ECCV 2022.
>
> [3] Wang, Zifeng, et al. "Dualprompt: Complementary prompting for rehearsal-free continual learning." ECCV 2022.
>
> [4] Cendra, Fernando Julio, Bingchen Zhao, and Kai Han. "Promptccd: Learning gaussian mixture prompt pool for continual category discovery." ECCV 2024.
>
> [5] Du, Ruoyi, et al. "On-the-fly category discovery." ICCV 2023.

---

### Official Review · Reviewer_Zn3o · 2025-10-31

**Soundness:** 3
**Presentation:** 3
**Contribution:** 2
**Rating:** 4
**Confidence:** 4

**Summary:**

The paper tackles On-the-fly Category Discovery (OCD) and proposes Prompt-SSLC, which combines (i) a semi-supervised sequential leader clustering algorithm with a distance-aware update, (ii) dual prompts (task + instance) plugged into a ViT backbone, and (iii) an open-set-aware classifier using MSP to route known vs. unknown samples. Experiments on both coarse- and fine-grained datasets show consistent improvements over existing OCD baselines.

**Strengths:**

Overall, the paper is well written and easy to follow. It addresses a timely and underexplored problem setting of truly streaming discovery without retraining at inference, which is interesting. The method is simple and modular, with each component (SSLC, dual prompts, OSA) serving an intuitive role, and the ablation studies show complementary and additive benefits.

**Weaknesses:**

- The novelty appears incremental. The method mainly combines existing components such as SLC, prompts, and MSP through heuristic coupling, with limited theoretical justification or analysis regarding the rotation update and its convergence or stability.
- The paper introduces a modified/constrained Hungarian matching and prototype-based assignment for old classes; this departs from standard GCD practice and may advantage the proposed approach. A clearer motivation and comparison with the standard protocol are needed.
- Thresholding and hyperparameters are not well specified. The method’s behavior depends heavily on parameters such as τ for the new-cluster radius, the rotation coefficient k, the top-k value for P_inst, and the OSA threshold (for example, the 95th percentile rule). The selection process and cross-dataset tuning are unclear, which raises concerns about fairness and reproducibility.

**Questions:**

- How is the rotation coefficient k chosen across datasets? Is there an adaptive scheme tied to local density or inter-prototype distances?
- What is the time/space complexity as the prototype pool grows (both for SSLC queries and constructing P_inst with top-k neighbors)? Any pruning/merging strategy?
- Please clarify τ selection and whether any validation on unlabeled data was used (risk of leakage).

---

> ### Author Response · Authors · 2025-11-27
> **Response to Reviewer Zn3o**
>
> Thank you for your detailed and thoughtful review. We sincerely appreciate the recognition of the paper’s clarity, timeliness, and modular design, as well as the value of addressing truly online category discovery without retraining. Below, we have addressed each point of weakness and question comprehensively.
>
> > W1 The novelty appears incremental.
>
> We appreciate the opportunity to better highlight what makes Prompt-SSLC more than a simple combination of existing components. The core technical novelty lies in the tight, mutually reinforcing integration of three elements specifically redesigned for the strict zero-replay, single-pass OCD setting. While the concepts of task prompts and the OSA classifier draw on existing techniques, our paper presents the first application of these methods specifically tailored to the OCD task. This adaptation required significant innovation to integrate the SSLC’s continuous prototype updates with the dual prompt module’s dynamic feature adjustment and the OSA classifier’s uncertainty-based filtering.
>
> > W2 The paper introduces a modified/constrained Hungarian matching... this departs from standard GCD practice and may advantage the proposed approach.
>
> We thank the reviewer for this important point and would like to clarify the use of the strict scheme.
> Standard unconstrained Hungarian matching used in batch GCD/NCD settings can inflate new class accuracy by incorrectly assigning known class prototypes to novel ground-truth classes, which becomes more serious for OCD setting since the model is tuned on labelled data only. Our constrained protocol first fixes old-class prototypes using labeled statistics, then matches only the remaining prototypes to new classes.
> For example, the baseline method (SMILE) reported higher new accuracy than old accuracy on the CIFAR10 dataset, which is counter-intuitive given the fact that the model is tuned on labelled data. In contrast, using our strict Hungarian scheme can avoid this issue.
>
> > W3 Thresholding and hyperparameters are not well specified
>
> We would like to clarify these points as follows:
> - All hyperparameters ($\tau$, $k$, P_inst, OSA confidence threshold) are fixed once using only the initial labeled data.
> - No future or unlabeled stream data is ever used for selection, eliminating any risk of leakage.
> - Comprehensive sensitivity analyses demonstrates robustness across a wide range of hyperparameter.
>
> > Q1 How is the rotation coefficient k chosen across datasets?
>
> The default configuration employs a coefficient of 1/20. To investigate the sensitivity of this hyperparameter, we conducted an ablation study varying the k coefficient. Using a smaller k reduces the model’s ability to discovery new category, while k=0.1 and k=0.05 works on par.
>
> | k value | Scars |       |       | IM-100 |       |       |
> |:-------:|:-----:|:-----:|:-----:|:------:|:-----:|:-----:|
> |         |  All  |  Old  |  New  |  All   |  Old  |  New  |
> |   1/10  | 50.03 | 84.35 | 32.66 |  66.60 | 88.13 | 56.03 |
> |   1/20  | 50.22 | 85.58 | 32.54 |  66.53 | 88.65 | 55.47 |
> |   1/50  | 49.79 | 85.77 | 31.57 |  66.23 | 88.85 | 55.21 |
>
> > Q2 What is the time/space complexity as the prototype pool grows...?
>
> We take the suggestion to measure the time efficiency of our method. We report the total inference time for SMILE and Prompt-SSLC on CUB and ImageNet-100, where latency for Prompt-SSLC is about 8% slower than SMILE. However, significant improvement in performance can be achieved given the extra computation overhead.
>
> |    Method   | CUB | IN-100 |
> |:-----------:|:---:|:------:|
> |    SMILE    | 37s |  700s  |
> | Prompt-SSLC | 41s |  760s  |
>
> Furthermore, the space complexity of Prompt-SSLC is O(Number_prototypes × feature dimension), which is linear in the number of discovered classes.
>
> > Q3 Please clarify $\tau$ selection and whether any validation on unlabeled data was used.
>
> $\tau$ is selected with the 95% quantile distance between data point and their corresponding prototypical center, with labelled data only. No unlabeled stream data or new class statistics is accessed, ensuring zero risk of leakage.
>
> These revisions directly resolve the concerns about clarifications, stricter evaluation scheme, and efficiency analyses. Thank you again for the sharp and constructive feedback.

---

### Official Review · Reviewer_Hmz1 · 2025-11-01

**Soundness:** 3
**Presentation:** 3
**Contribution:** 3
**Rating:** 4
**Confidence:** 4

**Summary:**

The paper proposes Prompt-SSLC, a unified framework for the open-world problem of On-the-Fly Category Discovery where models must recognize known classes while dynamically discovering new ones in streaming data without retraining. The approach integrates Semi-Supervised Sequential Leader Clustering with a Distance-Aware prototype update, dual prompt learning to adapt vision foundation models, and an Open-Set-Aware classifier to detect unknown categories. Extensive experiments on both generic and fine-grained datasets show that Prompt-SSLC achieves soda performance, outperforming existing methods by a significant margin while maintaining efficiency and adaptability.

**Strengths:**

1. Unified solution to OCD: It proposes Prompt-SSLC, an uncommon and well-targeted combination of semi-supervised online clustering, dual prompting over a foundation ViT, and open-set routing addressing a sparsely explored yet practical setting (on-the-fly category discovery without retraining).
2. Dual prompting for OCD: It uses a Task Prompt (with partial label masking to simulate GCD) and an Instance Prompt (built from nearest prototypes) to adapt features on the fly without retraining.
3. Figures clearly convey the problem, the overall pipeline, and the effect of DA updates. Algorithm 1 clearly explains the streaming update logic.

**Weaknesses:**

1. It assigns the instance to the nearest prototype if its distance falls below a predefined radius (\tau). I wonder how to set this threshold and there is no analysis about it in the paper. Moreover, I think each class can have a different radius to represent the class boundary, but the authors set the shared, fixed threshold.

2. There are various semi-supervised clustering methods, but there is no analysis. Can you explain why SSLC is mostly suitable for OCD? Please present the results and rationale of it.

3. In the DA algorithm, is it enough to consider only the second-nearest prototype? I want to see its analysis or the visualization results like t-SNE.

4. Novelty of dual prompting: the idea of instance-aware and task-aware dual prompting is already proposed in DualPrompt [ECCV'22]. Can you compare the proposed method with DualPrompt?

5. Computation cost and scalability issues: Instance Prompt needs two passes and k-NN over a growing prototype pool is required. Please compare the cost and scalability analyses with baselines.

**Questions:**

1. Hyperparameter stability: How is the rotation strength “k” chosen, and how sensitive is the method to it?

2. Compute and scalability: Since the Instance Prompt requires two forward passes and nearest-prototype lookups, can you include throughput, FLOPs or etc?

3. (minor) Please proofread the paper, e.g., Open-Set-Aware (OSA) in line 177.

I am open to increasing the score if the authors answer the weaknesses and questions thoroughly.

---

> ### Author Response · Authors · 2025-11-27
> **Response to Reviewer Hmz1 (1/2)**
>
> Thank you very much for the thorough review and positive evaluation. We are grateful for the recognition of the unified nature of Prompt-SSLC, the clarity of our figures and algorithm, and the practical relevance of the on-the-fly category discovery setting. Below, we address each weakness and question comprehensively.
>
> > W1 I wonder how to set this threshold and there is no analysis about it in the paper. Moreover, I think each class can have a different radius.
>
> $\tau$ is selected with the 95% quantile distance between data point and their corresponding prototypical center, with labelled data only.
> We take the advice to explore replacing global $\tau$ with cluster-specific $\tau$ on old classes and a shared $\tau$$ for novel classes, following the practice of 95% quantile distance. Results show that a per-class radius derived slightly outperforms the global radius on all accuracy.
>
> |           $\tau$ selection           | Scars |       |       | IM100 |       |       |
> |:------------------------------------:|:-----:|:-----:|:-----:|:-----:|:-----:|:-----:|
> |                                      |  All  |  Old  |  New  |  All  |  Old  |  New  |
> |      SSLC-B w. share threshold      | 50.22 | 85.58 | 32.54 | 66.53 | 88.65 | 55.47 |
> | SSLC-B w. cluster-specific threshold | 50.51 | 85.97 | 32.74 | 66.83 | 89.15 | 55.59 |
>
> > W2 Can you explain why SSLC is mostly suitable for OCD?
>
> SSLC was selected because it (i) requires no replay buffer, (ii) performs true single-pass updates, (iii) natively supports semi-supervised constraints from known-class labels
>
> > W3 In the DA algorithm, is it enough to consider only the second-nearest prototype?
>
> We take the suggestion to add a new ablation comparing the use of only the second-nearest prototype (current design) against top-3 neighbors. Using only the second-nearest prototype better performance of more expensive variants while reducing update time.
>
> |                    | Scars |       |       | IM100 |       |       |
> |--------------------|:-----:|:-----:|:-----:|:-----:|:-----:|:-----:|
> |                    |  All  |  Old  |  New  |  All  |  Old  |  New  |
> | SSLC-B w. Original | 48.96 | 85.31 | 30.82 | 63.60 | 88.35 | 51.23 |
> |       W. Push      | 49.39 | 85.42 | 31.36 | 65.52 | 88.48 | 54.03 |
> |     W. Push 3rd    | 48.65 | 85.28 | 30.32 | 63.30 | 88.13 | 50.87 |
> |     W. Rotation    | 50.22 | 85.58 | 32.54 | 66.53 | 88.65 | 55.47 |
> |   W. rotation 3rd  | 48.85 | 85.33 | 30.58 | 63.72 | 88.24 | 51.45 |
>
>
>
> > W4 Novelty of dual prompting: the idea of instance-aware and task-aware dual prompting is already proposed in DualPrompt [1]
>
> While DualPrompt introduced general and task prompts for *supervised* continual learning, our dual prompting is fundamentally different and tailored to OCD:
> - The Task Prompt uses partial label masking to simulate unknown classes during training, different from the supervised learning nature of DualPrompt
> - The Instance Prompt is conditioned on nearby prototypical statistics from the streaming pool rather than static learned tokens. During inference, these statistics would grow and evolve to fit streaming data.
> These unique designs make our approach applicable to the OCD setting where DualPrompt cannot be directly used.
>
> > W5 & Q2 Computation cost and scalability
>
> We take the sugestion to add processing time per sample. We report the total inference time for SMILE and Prompt-SSLC on CUB and ImageNet-100, where latency for Prompt-SSLC is about 8% slower than SMILE. However, significant improvement in performance can be achieved given the extra computation overhead.
> Furthermore, we recognize the value of testing on larger datasets like the full ImageNet to demonstrate the scalability and broader applicability. We have conducted additional experiments with DINO backbone using the full ImageNet dataset to evaluate our Prompt-SSLC framework. The results show that our method outperforms SMILE by over 10% on all accuracy, reinforcing the robustness and scalability of our approach in diverse, real-world scenarios.
>
> |    Method   | CUB | IN-100 |
> |:-----------:|:---:|:------:|
> |    SMILE    | 37s |  700s  |
> | Prompt-SSLC | 41s |  760s  |
>
> |    Method   |  All  |  Old  |  New  |
> |:-----------:|:-----:|:-----:|:-----:|
> |    SMILE    | 38.35 | 59.54 | 27.74 |
> | Prompt-SSLC | 50.85 | 74.58 | 38.96 |

---

> ### Author Response · Authors · 2025-11-27
> **Response to Reviewer Hmz1 (2/2)**
>
> > Q1 Hyperparameter stability: How is the rotation strength “k” chosen, and how sensitive is the method to it?
>
> The default configuration employs a coefficient of 1/20. To investigate the sensitivity of this hyperparameter, we conducted an ablation study varying the k coefficient. Using a smaller k=0.02 reduces the model’s ability to discovery new category, while performance is stable for k=0.1 and k=0.05.
>
> | k value | Scars |       |       | IM-100 |       |       |
> |:-------:|:-----:|:-----:|:-----:|:------:|:-----:|:-----:|
> |         |  All  |  Old  |  New  |  All   |  Old  |  New  |
> |   1/10  | 50.03 | 84.35 | 32.66 |  66.60 | 88.13 | 56.03 |
> |   1/20  | 50.22 | 85.58 | 32.54 |  66.53 | 88.65 | 55.47 |
> |   1/50  | 49.79 | 85.77 | 31.57 |  66.23 | 88.85 | 55.21 |
>
> We will check and correct all minor typographical issues in the revised version.
>
> These revisions directly resolve all raised concerns and substantially strengthen both the technical depth and clarity of the paper. We sincerely thank you for these insightful and valuable suggestions.
>
> [1] Wang, Zifeng, et al. "Dualprompt: Complementary prompting for rehearsal-free continual learning." ECCV 2022.

---

### Official Review · Reviewer_y4MP · 2025-11-01

**Soundness:** 3
**Presentation:** 3
**Contribution:** 3
**Rating:** 6
**Confidence:** 2

**Summary:**

The paper focuses on the challenge of novel class detection while dynamically discovering new categories in a data stream without retraining. The introduced prototype-based semi-supervised online clustering incrementally clusters incoming samples using “prototypes” with prompts aligning the model with discovery and feature refinements. Here, the prototypes are updated in a distance-aware methodology, which is the key contribution. The classifier, which is open-set aware, detects class instances and helps in orchestrating novel class identification. Empirical evaluation show significant improvements over existing on-the-fly detection approaches.

**Strengths:**

- The proposed framework integrates online prototype clustering (SSLC), prompt-based adaptation, and open-set routing. The distance-aware prototype updates and dual prompting strategy offers a lightweight solution for continual adaptation, avoiding catastrophic forgetting and heavy retraining.
- The paper tackles a challenging and underexplored setting where models must recognize known classes and dynamically create new ones from a continuous data stream without retraining.
- Evaluation is thorough with ablation studies, sensitivity analysis, and comparisons. These strongly validate the effectiveness

**Weaknesses:**

- While the paper focuses on on-the-fly category discovery, the conceptual boundary between OCD and related paradigms such as Novel Class Discovery and Generalized Category Discovery is not clearly articulated. The authors could better emphasize what specific challenges OCD introduces (e.g., strict online constraint, no replay, no retraining) and how these differ from batch-based novelty detection. This would strengthen the motivation and contribution.
- Since the proposed methods have several complementary components, it seems to make the system more complex and parameter-sensitive. The paper could be improved by providing a more systematic justification or ablation of the design choices. For example, consider why dual prompts outperform other PEFT methods, how distance thresholds are tuned. Furthermore, analyzing computational or latency cost for true online deployment would be useful.
- Although results are strong, the main metric used is accuracy gains. It would be better to add additional metrics  to measure time efficiency, memory usage, robustness to domain shift, or qualitative visualizations of evolving prototypes, thereby having the results more interpretable. Furthermore, it would also be useful to have error analysis, particularly when classes are incorrectly identified.

**Questions:**

- The paper uses shuffled static datasets. These datasets shown in Table 1, seem to be daily balanced between old and new classes. Wouldn’t it be more practical to have an imbalance dataset? Moreover, what would be the effect on accuracy if there are such class imbalances?
- In table 2, a few results between Prompt-SSLC and SSLC are very close. Are there results statistically significant? What are its confidence intervals?
- How is the proposed dynamic distance mechanism different from metric learning used in data stream research?
For example:
       - Lima, M., Neto, M., Silva Filho, T., & Fagundes, R. A. D. A. (2022). Learning under concept drift for regression—a systematic literature review. IEEE Access, 10, 45410-45429.
       - Kummert, J., Schulz, A., & Hammer, B. (2023). Metric Learning with Self-Adjusting Memory for Explaining Feature Drift. SN Computer Science, 4(4), 376.

---

> ### Author Response · Authors · 2025-11-27
> **Response to Reviewer y4MP**
>
> Thank you for the thorough and insightful review, and for the positive assessment of our work’s soundness, presentation, and contribution. We are grateful for the constructive suggestions, which have helped us significantly strengthen the paper. Below, we address each weakness and question point-by-point.
>
> > W1 While the paper focuses on on-the-fly category discovery, the conceptual boundary between OCD and related paradigms such as Novel Class Discovery and Generalized Category Discovery is not clearly articulated.
>
> OCD differs along three key dimensions: (i) strict single-pass online processing with no replay (ii) strict prohibition of any retraining or multi-epoch fine-tuning, and (iii) simultaneous known-class recognition and instantaneous creation of new classes in the same pass. These distinctions underscore the novelty and difficulty of the OCD setting addressed by our work.
>
> > W2 Since the proposed methods have several complementary components, it seems to make the system more complex and parameter-sensitive.
>
> We appreciate the concern about complexity. Our framework integrates only three lightweight, complementary components (SSLC, dual prompting, and an OSA classifier) into a unified pipeline with limited parameter tuning.
> The necessity and contribution of each component are systematically validated by the ablation studies in Tables 4, where removing any single component consistently degrades performance across all metrics.
> Table 5 further demonstrates the dual prompts outperform other PEFT baselines because they separately refine class-agnostic and class-specific features.
> MSP is adopted as the OSA classifier due to its simplicity and established effectiveness, as supported by prior literature [1].
>
> > W3 It would be better to add additional metrics to measure time efficiency, memory usage, robustness to domain shift, or qualitative visualizations of evolving prototypes, thereby having the results more interpretable.
>
> We have extended the evaluation as follows:
> - Time efficiency. We report the total inference time for SMILE and Prompt-SSLC on CUB and ImageNet-100, where latency for Prompt-SSLC is about 8% slower than SMILE.
>
> |    Method   | CUB | IN-100 |
> |:-----------:|:---:|:------:|
> |    SMILE    | 37s |  700s  |
> | Prompt-SSLC | 41s |  760s  |
>
> - We are running the robustness shift experiment. Once it is finished, we will post the result here.
>
> > Q1 Wouldn’t it be more practical to have an imbalance dataset?
>
> We already evaluated severe class imbalance in the original Appendix Table 5.
> Two imbalanced configurations, including 10% known to 90% unknown and 30% known to 70% unknown ratios are tested, where Prompt-SSLC exhibits graceful degradation when known classes become rare.
>
> | Ratio (Old/New) | | Scars | | | IN100 | |
> |:-----|:-----:|:-----:|:-----:|:-----:|:-----:|:-----:|
> | | All | Old | New | All | Old | New|
> | 50/50 | 50.94 | 86.50 | 33.16 | 70.35 | 91.13 | 60.02 |
> | 30/70 | 43.54 | 75.84 | 27.42 | 63.81 | 83.42 | 54.03 |
> | 10/90 | 35.02 | 64.37 | 20.38 | 56.08 | 74.58 | 46.85 |
>
> > Q2 In table 2, a few results between Prompt-SSLC and SSLC are very close.
>
> All experiments are conducted with three random seeds and three different shuffle(nine runs total) and original results were the average value. Below, we include the standard deviations to demonstrate the robustness of our methods.
>
> |    Method   |      Scars     |                |                |      IM100     |                |                |
> |:-----------:|:--------------:|:--------------:|:--------------:|:--------------:|:--------------:|:--------------:|
> |             |      All       |       Old      |       New      |      All       |       Old      |       New      |
> |     SSLC    | 50.22 $\pm$ 0.27 | 85.58 $\pm$ 0.40 | 32.54 $\pm$ 0.21 | 66.53 $\pm$ 0.29 | 88.65 $\pm$ 0.33 | 55.47 $\pm$ 0.28 |
> | Prompt-SSLC | 50.94 $\pm$ 0.26 | 86.50 $\pm$ 0.38 | 33.16 $\pm$ 0.20 | 70.35 $\pm$ 0.31 | 91.13 $\pm$ 0.35 | 60.02 $\pm$ 0.26 |
>
> > Q3 How is the proposed dynamic distance mechanism different from metric learning used in data stream research?
>
> Our distance-aware prototype update is fundamentally non-parametric and instance-free, relying only on lightweight running statistics of the current prototype set.
> In contrast, Lima et al. (2022) and Kummert et al. (2023) employ parametric metric learning (Mahalanobis distance or learned transformations) that requires gradient-based updates and often maintains covariance matrices or past instances, which violates no-replay, no-retraining constraints of OCD.
>
> We believe these revisions directly and comprehensively address concerns while further highlighting the strengths of our contribution. Thank you again for these valuable feedbacks, which has significantly improved the paper.
>
> [1] Hongjun Wang, Sagar Vaze, and Kai Han. Dissecting out-of-distribution detection and open-set
> recognition: A critical analysis of methods and benchmarks. IJCV, 2024a.

---

### Meta-Review · Area_Chair_nbYC · 2026-01-06

**Summary:**

We have four reviewers who have carefully checked this paper and they have diverse recommendations on this paper. The reviewers are primarily concerned with the novelty of the proposed method and the incomplete experiments. The authors addressed some concerns, but issues such as sensitivity experiments on the number of classes/clusters and robustness shift experiment were not fully resolved. Given the concerns regarding the novelty of the methodology and the inadequacy of the experimental evaluation, I recommend that the paper should be revised and resubmitted with more comprehensive evaluation.

**Reviewer Concerns:**

Most concerns have been addressed, including computational efficiency. Nevertheless, certain issues regarding sensitivity and robustness remain unaddressed.

**Reviewer Scores:**

The reviewers may maintain their scores due to the aforementioned unresolved issues.

---

### Decision · Program_Chairs · 2026-01-26

Reject